# Neuregulin (NRG-1β) Is Pro-Myogenic and Anti-Cachectic in Respiratory Muscles of Post-Myocardial Infarcted Swine

**DOI:** 10.3390/biology11050682

**Published:** 2022-04-29

**Authors:** Cristi L. Galindo, Van Thuan Nguyen, Braxton Hill, Ethan Easterday, John H. Cleator, Douglas B. Sawyer

**Affiliations:** 1Department of Biology, Ogden College of Science & Engineering, Western Kentucky University, Bowling Green, KY 42101, USA; vanthuan.nguyen@wku.edu (V.T.N.); braxton.hill843@topper.wku.edu (B.H.); ethan.easterday493@topper.wku.edu (E.E.); 2Centennial Heart at Skyline, 3443 Dickerson Pike, Suite 430, Nashville, TN 37207, USA; john.cleator@hcahealthcare.com; 3Department of Cardiac Services, Maine Medical Center, Scarborough, ME 04074, USA

**Keywords:** myocardial infarction, neuregulin, glial growth factor 2, pre-clinical therapy, skeletal muscle, gene expression, RNA sequencing

## Abstract

**Simple Summary:**

Neuregulin is a growth factor that has been shown to prevent adverse remodeling in the heart and may represent a therapeutic for patients with systolic heart failure. A common symptom in heart failure is shortness of breath, which has been related in part to impaired skeletal muscle function. Since neuregulin directly activates skeletal muscle, in addition to heart tissue, we hypothesized that neuregulin might directly affect intercostal muscle gene expression changes in heart disease. We tested this hypothesis by performing global gene expression analysis of intercostal muscle tissue collected from pigs treated with recombinant neuregulin after the induction of myocardial infarction, an experimental model clinically similar to a human heart attack. We found that neuregulin-treated pigs had massive changes in global gene expression consistent with new muscle cell formation, as compared to untreated pigs. These data suggest that neuregulin is an important mediator of muscle function that can potentially be used to study heart disease-associated muscle dysfunction and the development of new therapeutics aimed at muscle repair in heart failure, as well as other diseases associated with muscle dysfunction and weakness.

**Abstract:**

Neuregulin-1β (NRG-1β) is a growth and differentiation factor with pleiotropic systemic effects. Because NRG-1β has therapeutic potential for heart failure and has known growth effects in skeletal muscle, we hypothesized that it might affect heart failure-associated cachexia, a severe co-morbidity characterized by a loss of muscle mass. We therefore assessed NRG-1β’s effect on intercostal skeletal muscle gene expression in a swine model of heart failure using recombinant glial growth factor 2 (USAN-cimaglermin alfa), a version of NRG-1β that has been tested in humans with systolic heart failure. Animals received one of two intravenous doses (0.67 or 2 mg/kg) of NRG-1β bi-weekly for 4 weeks, beginning one week after infarct. Based on paired-end RNA sequencing, NRG-1β treatment altered the intercostal muscle gene expression of 581 transcripts, including genes required for myofiber growth, maintenance and survival, such as MYH3, MYHC, MYL6B, KY and HES1. Importantly, NRG-1β altered the directionality of at least 85 genes associated with cachexia, including myostatin, which negatively regulates myoblast differentiation by down-regulating MyoD expression. Consistent with this, MyoD was increased in NRG-1β-treated animals. In vitro experiments with myoblast cell lines confirmed that NRG-1β induces ERBB-dependent differentiation. These findings suggest a NRG-1β-mediated anti-atrophic, anti-cachexia effect that may provide additional benefits to this potential therapy in heart failure.

## 1. Introduction

Chronic heart failure (CHF) is a complex clinical syndrome resulting from any disorder that impairs cardiac function such as ventricular filling or the ejection of blood into the systemic circulation. Impaired muscle formation can be a cause of several muscle-related [1] diseases as well as age-related muscle deterioration, also known as sarcopenia [2,3]. Although the precise mechanisms are not fully understood, some studies recently showed that systemic oxidative stress [4], exercise intolerance [5], a low grade of inflammation, abnormal energy metabolism, transition of myofibers, mitochondrial dysfunction, a reduction in muscular strength, and muscle atrophy play an important role in skeletal muscle dysfunction/abnormalities in the setting of CHF [6,7]. However, abnormalities in skeletal muscle metabolism, total muscle mass, and peak functional capacity in patients with CHF cannot be fully explained by lowered cardiac output [8] or inadequate skeletal muscle oxygenation [9].

The neuregulin (NRG) family is a large class of neuronal growth factors that induce signaling via type I epidermal growth factor receptor (EGFR) tyrosine kinases (ErbB2, ErbB3, and ErbB4). NRG-1 is essential for the development of the sympathetic nervous system [10] and is required for the formation and maintenance of neuromuscular synapses. In postembryonic skeletal muscles, local production of NRG-1 by motor neurons increases acetylcholine receptor expression and accumulation at motor nerve terminals [11]. Several NRG-1 isoforms, including NRG-1α, NRG-1β Type I, and NRG-1β type II (also called glial growth factor 2, GGF2), were shown to stimulate myoblast differentiation in immortalized and primary myoblasts [12,13,14]. Additionally, ErbB3 is induced during muscle cell differentiation [15] and regulates muscle progenitor cell fate and thus the balance between stem cell renewal and differentiation [16].

We previously reported results from a pre-clinical trial of the effects of a pharmaceutical grade of NRG-1/GGF2 (cimaglermin alfa) using a swine model of heart failure. In this previously published study [17], a balloon was inserted into the descending coronary artery of pigs to induce myocardial infarction. GGF2 was administrated intravenously twice weekly, starting at 1 week after infarction. Tissues were collected 35 days thereafter, and left ventricular tissue remote from the infarct was interrogated for global gene expression using microarrays. GGF2-treated animals exhibited better cardiac function, including less maladaptive myocardial remodeling, reduced fibrosis, and gene expression changes consistent with better cardiomyocyte survival but less myofibroblast-driven extracellular matrix remodeling. In the study presented here, we performed deep sequencing on intercostal muscles of these same pigs to determine the effects of GGF2 treatment on post-MI skeletal muscle function. We also compared these results with GGF2-mediated expression changes in cardiac muscle and myopathic genes obtained by analyzing repository expression data. As detailed herein, our results support a role for exogenously delivered GGF2 in skeletal muscle remodeling and repair after cardiac injury.

## 2. Materials and Methods

Animals: Intercostal muscle samples were acquired from swine that were used to test the efficacy of NRG treatment on swine after myocardial infarction, the details of which were previously published [17]. Briefly, eight pigs underwent balloon occlusion surgery to introduce a left ventricular infarct. At one week post-MI, five of the animals received biweekly intravenous injections of GGF2 (three at a dose of 0.67 mg/kg and two at 2 mg/kg). The remaining swine served as untreated controls. As reported previously [17,18,19], high-dose treatment was associated with symptomatic hypoglycemia. In consultation with veterinary staff and IACUC, dose adjustment was made for the remaining swine in the treatment group.

Sequencing: Total RNA was isolated using RNeasy Mini kit (Qiagen), following the manufacturer’s instructions, RNA integrity was confirmed using Agilent Bioanalyzer, and RNA sequencing was performed by the Vanderbilt Technologies for Advanced Genomics core on a total of eight animals, all of which received myocardial infarction as previously described [17]. RNA libraries were constructed using the Illumina TruSeq Stranded Total RNA kit with Ribo-Zero Gold. Samples were sequenced on the Illumina HiSeq 2500 using v3 SBS chemistry. Libraries were sequenced on paired-end-50 flow cell runs at ~35 M PF reads per sample.

Data analysis: Sequences were aligned to the susScr3 assembly of the pig genome and analyzed using the Tophat2 + CuffDiff platform provided by DNAnexus, Inc. (Mountain View, CA, USA). Genes were annotated using the Ensemble, UCSC, and NCBI iGenomes datasets. Partek Genomics Suite 6.6 was also used to perform statistical analyses of aligned sequences, as well as hierarchical clustering and principal components analysis (PCA). Differential sequences that were determined to be statistically significant (multiple hypothesis-corrected *p*-value < 0.05 and fold-difference > 2.0) among the three different groups of animals were compiled into a single list and the annotations were manually verified using information from UCSC, Ensemble, and NCBI databases. For uncharacterized transcripts, the differential Sus scrofa sequences were queried against the human reference genome using the BLAT tool provided on the UCSC Genome Browser website to identify orthologous human genes. Uncharacterized transcripts were included in the final list if they were at least 75% similar to an annotated human gene.

Functional enrichment analysis with Bonferroni multiple hypothesis adjustment was performed using GSEA, available online from the BROAD Institute. Individual gene functions were identified using public databases and literature reports. Functional subgroups of genes that were detected by exhaustive, manual annotation could generally be recapitulated using statistical functional analysis tools but were missed due to lower significance relative to hundreds of other, largely redundant ontological categories (e.g., “tissue development”). To better identify these potentially relevant functional groups while minimizing user bias, we also organized genes hierarchically according to their biological processes within the Gene Ontology Consortium classification system. The top three non-redundant biological processes were chosen based on the numbers of differential genes included and functional specificity, whereby non-informative categories, such as “Regulation of Cell Process”, were ignored. A divisive hierarchical clustering approach was applied for consecutive subgrouping. All ontologies generated from the gene lists, irrespective of statistical significance, were examined to gain a complete functional overview; however, a statistically significant overlap (Bonferroni-adjusted *p*-value < 0.05) was required for inclusion in the subsequently generated functional hierarchy reported herein.

Cell Culture: C2C12 cells and L6 myoblasts were cultured in growth media (GM) consisting of Dulbecco’s modified Eagle’s medium (DMEM) (25 mM glucose) supplemented with 10% fetal bovine serum (FBS) and 1% antibiotics (penicillin + streptomycin). GM was changed every alternate day until cells were 90% confluent. GM was then changed to differentiation media (DM) consisting of DMEM (25 mM glucose) supplemented with 2% horse serum and 1% antibiotics. Compounds or vehicle controls were added at Day 0 (i.e., the time at which the cells were switched from GM to DM). The additional methodology is provided in Appendix A.

Quantification of myoblast fusion and myotube numbers. C2C12 or L6 cells were differentiated according to the conditions of described experiments and fixed in 4% paraformaldehyde for 25 min. Next, the cells were stained with Jenner–Giemsa dyes as described in (Velica). The multiple microphotographs of stained cells were taken using an inverted microscope, and at least 4 different fields were used to estimate the total number of cells and the number of myotubes. The cells with more than 3 nuclei were counted as the number of myotubes. The fusion index was determined as the number of cells with more than 3 nuclei per cell divided by the total number of cells. In some experiments, the myotube numbers were estimated without the cell staining based on microphotographs of growing cells.

Western blot and immunoprecipitation analysis. The cells were solubilized by scraping into the cold NP-40 lysis buffer (10 mM Tris-HCl, pH 7.5, 150 mM NaCl, 1% Nonidet P-40 containing protease inhibitors, and 1 mM Na3VO4). The lysates were then clarified by centrifugation (14,000× *g*, 10 min). Aliquots containing equal amounts of protein were subjected to 4–20% gradient SDS-PAGE. Subsequently, proteins were transferred to nitrocellulose membranes, and the membrane was blocked by incubation with Odyssey blocking buffer (LI-COR) for 1 h at room temperature. The membrane was then incubated overnight at 4 °C with the indicated antibody in the same blocking solution washed 3 times in the TBS buffer containing 0.05% Tween 20 (TBSTw) and incubated for 1 h with HRP-conjugated secondary antibody in SEA BLOCK blocking buffer (Thermo Fisher, Waltham, MA, USA). The membranes were then washed three times with the TBSTw buffer and visualized using the Fujifilm LAS-4000 imaging system. For immunoprecipitation, the cells were grown and differentiated in 10 cm tissue culture dishes and lysed in 0.5 mL of cold NP-40 lysis buffer. The lysates were precleared by incubation with 100 μL of protein A agarose beads (Santa-Cruz, Dallas, TX, USA) for 2 h at 4 °C and then incubated with 5 µg of C-18 antibodies against ErbB-2 and 50 μL of protein A agarose overnight at 4 °C. The beads were washed 3 times in TBSTw, then boiled in 1× Laemmli sample buffer for 2 min, and subjected to Western blot.

## 3. Results

### 3.1. GGF2 Alters Intercostal Muscle Gene Expression

Comparison of GGF2-treated and untreated pigs yielded a total of 581 differentially expressed transcripts including 500 annotated swine genes and 81 uncharacterized transcripts with at least 75% similarity to an annotated human gene (Appendix A). Not surprisingly, there were nearly twice as many transcriptional changes in animals treated with the high dose of GGF2 (512 genes) compared to the post-MI swine that received the lower dose (311 genes) versus untreated controls. These dose-dependent differences resulted in a clear separation of the two groups from one another, secondarily to untreated samples, upon hierarchical clustering of standardized hybridization values (Figure 1a). Of the 242 genes differentially expressed between treated and untreated swine at both doses (Figure 1b), only one transcript was altered in disparate directions compared to controls (MYLIP, down-regulated 1.7-fold in low dose-treated and up-regulated two-fold in high dose-treated animals, Appendix A). Of the remaining 241 similarly altered genes, 91 and 150 were up and down-regulated in treated animals, respectively (Appendix A).

### 3.2. Dose-Dependent Effects of GGF2

More than half of the genes that were altered in both low dose and high dose GGF2-treated swine (133 out of 242) exhibited a “dose-like” expression pattern (61 up-regulated and 72 down-regulated transcripts). Notable examples of up-regulated genes were those important for muscle cell development and contraction, as well as genes involved in amino acid and carbohydrate metabolism. Down-regulated genes displaying a “dose-like” response included those involved in lipid metabolism and insulin signaling, ECM structure and maintenance, and regulation of vascular functions. Notable dose-specific genes included those important for myofiber growth and muscle growth maintenance, such as slow-twitch MYL6B (down-regulated) and myosin heavy chain genes MYH3 and MYHC (up-regulated). The gene encoding ladybird homeobox 1 (LBX1), which is a key regulator of migratory muscle-specific stem cell precursors required for the development of forelimb muscles, including the diaphragm, was also up-regulated (Figure 2).

### 3.3. Functional Enrichment of GGF2 Altered Genes

Although many more genes were altered in post-MI pigs that received the higher dose than those treated with low dose GGF2, enriched functions were similar. The most significantly over-represented biological processes for both gene lists were tissue development, cellular proliferation and differentiation, organ morphogenesis, and ECM organization (Figure 3a). Functional analysis of the larger gene set (500+ genes altered in either of the two doses) resulted in nearly identical results. The gene lists were therefore combined for further analyses and the average fold-change and *p*-values were used where applicable.

To identify relevant functional subgroups of GGF2-altered genes and thereby obtain more specific mechanistic information, genes were hierarchically ordered based on their official Gene Ontology Consortium classifications. Based on this heuristic approach, the top non-redundant biological processes encompassing 445 out of all 581 differentially expressed genes were (1) developmental process (267 genes, *p* = 1 × 10^−21^, enrichment score (ES) = 1.8), (2) regulation of primary metabolic process (208 genes, *p* = 1.8 × 10^−3^, ES = 1.4), and (3) response to stress (139 genes, *p* = 1 × 10^−2^, ES = 1.5). Some genes were classified in more than one of these categories and thus were “counted” more than once for subsequent functional subdivisions (i.e., 267 + 208 + 139 = 614 ≠ 445 genes). Developmental genes were further parsed by tissue type (neurogenesis, vascular, muscular, ECM, and adipose), as were metabolic and stress response genes (Figure 3b).

Notably, more than two dozen GGF2-altered genes have known roles in fat cell metabolism or signaling. For example, adiponectin (ADIPOQ), predicted gene with 89.7% similarity to human adiponectin receptor 1 (ADIPOR1), predicted gene with 83.7% similarity to human adipogenesis regulatory factor (ADIRF), CCAAT/enhancer binding protein (C/EBP), alpha (CEBPA), perilipin (PLIN1) and peroxisome proliferator-activated receptor gamma (PPARG), which were all down-regulated (2.9–28.2-fold, Appendix A) encode markers of adipogenesis. Consistent with lowered adipogenesis, genes encoding lipid sensors and fatty acid metabolism were altered, as were genes associated with carbohydrate metabolism and insulin signaling. ECM genes were also mainly down-regulated, whereas muscle-associated transcripts were generally up-regulated (Appendix A).

### 3.4. Genes Altered in Skeletal and Cardiac Muscles of GGF2-Treated Post-MI Pigs

We previously analyzed gene expression in left ventricular tissues remote from infarct in these same animals using *Sus scrofa* microarrays [17]. This prior study indicated that GGF2 treatments reduced profibrotic transcripts, lowered percentages of activated fibroblasts, and inhibited cardiac fibrosis [17]. We thus compared our prior ventricular data to transcriptional changes in intercostal skeletal muscle tissues to identify potentially “global” GGF2-mediated alterations. As shown in Table 1, there were 32 genes that were similarly altered in skeletal and cardiac muscle of GGF2-treated post-MI pigs, 18 of which were previously shown to be functionally relevant specifically in skeletal muscle and/or the cardiovascular system [20,21,22,23,24,25,26,27,28,29,30,31,32,33,34,35,36,37,38,39,40,41,42,43,44,45,46,47,48,49,50,51,52,53]. To determine whether these commonly altered genes are functionally associated, we performed protein–protein interaction network functional enrichment analysis, using the publicly available online tool, STRING. For context, we also included NRG1 and relevant ErbB receptors (ErbB2-4). The resulting network included 24 of the input proteins, with three subgroups based on K means clustering (Figure 4).

### 3.5. Comparison to Skeletal Muscle Myopathies

GGF2-altered genes important for muscle development included those that have been previously implicated in myocyte cell dysfunction, gross alterations in muscle mass, and heart disease (Table 1). We therefore compared GGF2-altered genes to four animal models of skeletal myopathy and 11 human myopathies (Table 2) and found 230 cachexia transcripts reversibly altered in GGF2-treated pigs (e.g., up-regulated in myopathic skeletal muscle and down-regulated in GGF2-treated pigs). The greatest overlap among experimental animal studies was the mouse denervation model of myopathy (70 genes), followed by cardiac cachexia (46 genes) and 24 h starvation (31 genes) (Figure 5a). Conversely, only five of the genes differentially expressed in rats with experimental Type I diabetes, compared to non-diabetic controls, were altered in the reverse direction in GGF2-treated versus untreated pigs. A similar situation was observed for Type II diabetes in humans (only 8 overlapping genes). Interestingly, the greatest overlap observed for human diseases was for Pompe disease (63 genes), followed closely by tibial muscular dystrophy (55 genes).

Notable examples of overlapping transcripts included the gene encoding thrombospondin 4 (THBS4), which was down-regulated 2.6-fold in GGF2-treated swine (*p* = 0.001) and up-regulated in denervated animal muscles (1.6-fold, *p* = 0.048) and in humans with Pompe (2-fold, *p* = 0.046), tibial muscular dystrophy (3.1-fold, *p* = 0.011), iron-sulfur cluster myopathy (2-fold, *p* = 0.046) and old age (1.6-fold, *p* = 0.039). Other myotrophic genes that were reversibly down-regulated in response to GGF2 treatment were CHRNA1, which encodes the alpha subunit of the nicotinic cholinergic receptor, syndecan 4 (SDC4), dermatopontin (DPT), the stem cell marker KIT, the cardiac muscle myofibrillar stretch-sensor ankyrin repeat domain 1 (ANKRD1), matrix metalloproteinase 2 (MMP2) and a negative regulator of insulin secretion (GPR137B). GGF2 also induced the expression of genes that are down-regulated in humans and/or animal experimental models of myopathy, including KLF10 (encodes Kruppel-like factor 10), purinergic receptor P2Y (P2RY1), the gene encoding the lactate transporter MCT4 (SLC16A3), retinoid receptor γ (RXRG), triadin (TRDN), myosin heavy chain (MYH1), epidermal growth factor (EGF), and the skeletal muscle isoform of phosphorylase kinase (PHKA1) that when mutated causes muscle glycogenosis (Figure 5b).

### 3.6. GGF2/NRG-1β Stimulates ErbB2-Dependent Myoblast Differentiation

To validate NRG-induced myogenesis, immature L6 or C2C12 myoblasts were cultured in differentiation media and treated with GGF2 or recombinant human NRG-1β for 5–6 days. Consistent with previous studies [12,13,81], NRG-1β and GGF2 enhanced myoblast fusion and myotube formation (Figure 6a and Appendix A). GGF2/NRG-1β-induced muscle cell differentiation was accompanied by increased protein levels of differentiation markers, including myosin heavy chain (MHC) and myotubule (M)-cadherin, as well as phosphorylated AKT, glycogen synthase kinase 3-α (GSK-3α), and focal adhesion kinase (FAK) (Figure 6b and Appendix A). ErbB2 expression and phosphorylation were increased during differentiation (Appendix A), and differentiation was abrogated by the addition of the ErbB2 inhibitor TAK165 (Figure 6a), as well as by other pharmacological inhibitors or siRNA (Appendix A).

### 3.7. A Putative GGF2-Induced Skeletal Muscle Signaling Pathway

Based on gene set enrichment analysis (GSEA), top transcription factors for GGF2-altered genes included the pro-adipogenic factor CCAAT enhancer binding protein β (C/EBP-β, 101 genes, FDR q value = 3.8 × 10^−33^), as well as the myogenic differentiation factors myoblast determination protein 1 (MYOD, 19 genes, FDR q value = 5.4 × 10^−9^) and myocyte enhancer factor-2 (MEF2, 21 genes, FDR q value = 1.1 × 10^−8^). A more comprehensive functional analysis using Ingenuity Pathway Analysis (IPA) software likewise identified these same three factors, as well as all three relevant ERBB receptors (Table 3), suggesting that more than one skeletal muscle cell type contributed to the observed transcriptional changes. Similarly, 45 genes were altered downstream of fibroblast growth factor (FGF2, *p* = 4.8 × 10^−20^, Table 3), which is a known regulator of skeletal muscle proliferation and differentiation and additionally influences intramuscular adiposity by regulating trans-differentiation of fibro-adipogenic progenitors [82].

In addition to providing significance (*p*-value), IPA functional analysis additionally predicts directionality for some upstream factors (Table 3), based on known regulatory information derived from the scientific literature and other knowledge repositories. This information, along with manual searches of differentially expressed genes using PubMed and various online gene, protein, and signaling pathway databases, was used to construct a potentially meaningful functional network (Figure 7). This heuristic approach resulted in a clear pattern of down-regulated versus up-regulated genes, proteins, and predicted regulatory factors which generally correlated to cell types. Using these results as a guide, cell-type-specific proliferation and differentiation was inferred beginning with mesenchymal stem cells (MSC), which normally proliferate in response to platelet derived growth factor-BB (PDGF BB, 36 downstream genes altered, *p* = 3.7 × 10^−22^, activation Z score = 1.6, Table 3) and epidermal growth factor (EGF, transcriptionally up-regulated 3.5-fold, Appendix A and 49 downstream genes affected, Table 3) and differentiate into fibro-adipogenic progenitors or myoblasts in response to platelet-derived growth factor receptor alpha (PDGFRA, transcriptionally downregulated -2.1-fold, Appendix A) and ladybird homeobox 1 (LBX1, transcriptionally upregulated 3.4-fold, Appendix A), respectively.

Concomitantly, myogenesis inhibitor Twist1 was also predicted as inhibited (Z = −2.4, Table 3). Twist 1 maintains skeletal muscle progenitors, in part by inhibiting transactivation of myocyte enhancer factor 2C (MEF2C), which was predicted as being activated in GGF2-treated post-MI pig skeletal muscle (Z = 2.9, Table 3). Continuing this process revealed generalized decreases in adipocyte and fibroblast lineages (blue boxes, Figure 7), along with increased MSCs and myoblast lineages (red boxes, Figure 7).

## 4. Discussion

The results presented herein show that exogenous delivery of a pharmaceutical-grade version of NRG/GGF2 not only improves post-MI cardiac function but may also promote better intercostal skeletal muscle function via shifting the balance of stem-cell derived differentiated cell types. The global expression profiles exhibited in muscles of treated post-MI pigs were overwhelmingly consistent with fewer adipocytes and myofibroblasts and conversely with higher numbers of myofibers. The combined methodologies comparing transcript levels with basic functional analyses, a meta-analysis of previously published muscle studies, and manual review of public databases and scientific literature provided a surprisingly complete and plausible molecular signature relevant to NRG-mediated multi-organ responses to cardiac injury.

In addition to predictive alterations in the activity levels of local transcription factors and intracellular signaling molecules, the functional analysis identified multiple circulating factors that could account for some of the observed transcriptional changes in skeletal muscle. Inhibition of TGF-β1, for example, might reflect systemic changes given that TGF-β is generally increased with chronic inflammation and has been proposed as a biomarker of coronary artery disease in the setting of acute heart failure [83]. Local skeletal muscle responses to lower circulating levels of AGT might likewise result from GGF2-mediated alterations in blood volume and blood pressure within the cardiovascular system. In support of this, GGF2 treatment altered several genes consistent with effects of the nonsteroidal anti-inflammatory drug clopidogrel (11 genes, *p* = 4.6 × 10^−7^, Z = 3.302, Table 3) and the hypertension drug enalapril (13 genes, *p* = 9.3 × 10^−11^, Z = 3.073, Table 3). The SERCA2 inhibitor thapsigargin was similarly identified based on the directionality and expression of a cluster of 20 genes (6.5 × 10^−8^, Z = 3.450), suggesting that in skeletal muscle GGF2 mimics some of the cardioprotective actions associated with commonly used heart medications, possibly involving prolonged calcium-induced contractions and increased blood flow.

Although this study did not include sequencing of small RNAs, several micro-RNAs were nonetheless predicted as being present at the local tissue level or in the circulation. For instance, miR-29b-3p was predicted as being activated based on functional analysis (Z = 3.1, Table 3), which is notable, because miR-29b-3p promotes muscle differentiation by reducing myoblast proliferation and inducing myotube formation [84,85]. Similarly predicted as activated (Z = 2.8, Table 3), miR-335-3p was reported to play a role in muscle regeneration, influencing both myoblast differentiation and fiber type transformation [86,87]. On the other hand, sarcopenia-associated miR-21 [88,89] was predicted to be inhibited, based on the alteration of 17 mRNA targets in post-MI pigs (Table), which is of interest because mir-21 was recently reported to be a circulating biomarker for accelerated sarcopenia in patients with chronic heart failure [88]. GGF2 might also influence heart failure-associated muscle dysfunction via increased mir-30c-5p, which might represent a therapeutic target based on protection against myocardial ischemia-reperfusion injury in rats [90] and decreased plasma levels in patients with coronary heart disease [91].

Considered together, the data presented here strongly support the role of NRG/GGF2 in skeletal muscle maintenance and turnover in general and especially in the setting of cardiac injury and heart failure. Systemic treatments with circulating growth facts, such as NRG, may be problematic due to diverse effects, given the importance of NRG in multiple interrelated systems. On the other hand, these types of growth factor-based treatment have the benefit of potentially correcting abnormalities in multiple organs. In this study, we provide insights into the underlying molecular mechanisms of NRG-mediated effects inducible by cardiac injury-mediated stress in an “unrelated” tissue that is affected secondarily by heart dysfunction. Future studies will be aimed at dissecting cell-level effects of NRG/GGF2 and further investigating those identified downstream factors that might represent putative and highly specific therapeutic candidates for other muscle-involved disorders. In addition, future investigation examining respiratory muscle function will be needed to understand the potential clinical benefits of NRG/GGF2′s actions in skeletal muscle in this context.

## 5. Patents

There are no patents associated with this study.

## Figures and Tables

**Figure 1 biology-11-00682-f001:**
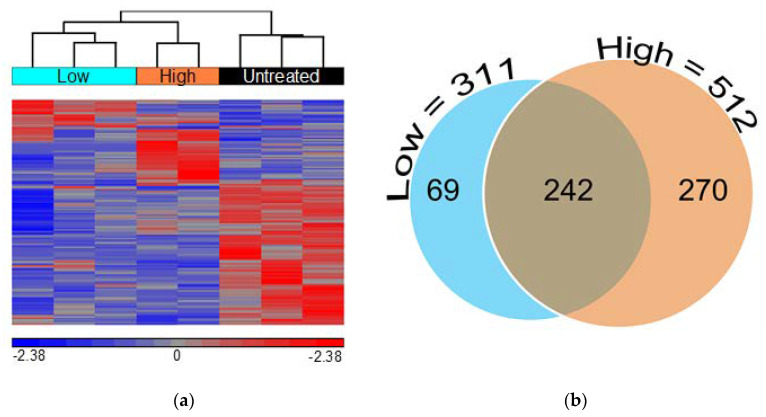
Low and high doses of GGF2 alter intercostal skeletal muscle transcription. (**a**) Hierarchical clustering of 581 genes significantly differentially expressed between GGF2-treated and untreated post-MI swine. Columns represent individual samples for each group, and rows represent individual probes/genes. Red, blue, and gray represent the highest, lowest, and medial fluorescent signal values, respectively. (**b**) Venn diagram showing numbers of genes differentially expressed in post-MI swine treated with GGF2 at low (teal circle, 311 genes) and high doses (orange circle, 512 genes) compared to untreated control animals.

**Figure 2 biology-11-00682-f002:**
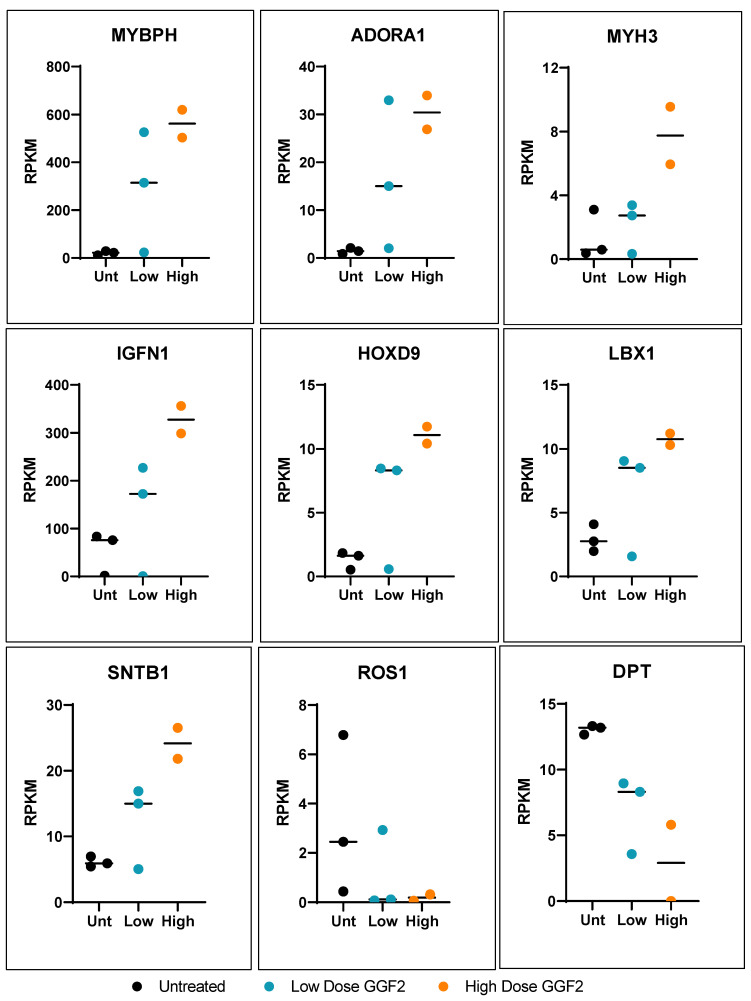
Dose-like response genes altered in GGF-2 treated pigs post-MI. Dot plots show relative counts as reads per kilobase per transcript, per million mapped reads (RPKM) for genes exhibiting fold-differences for GGF2 versus untreated pigs (*y*-axis). Individual samples are indicated on the *x*-axis and by color. Black represents untreated (Unt), teal represents low-dose (0.67 mg/kg/day), and high dose (2 mg/kg/day) is shown in orange.

**Figure 3 biology-11-00682-f003:**
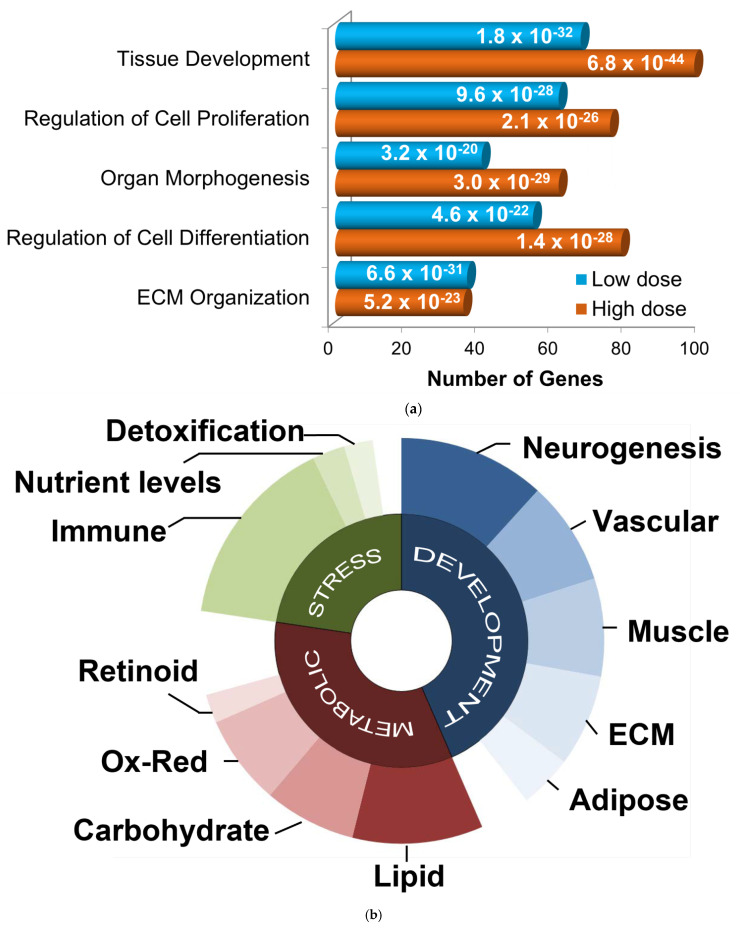
Functional categorization of GGF2-altered genes. (**a**) Bar chart showing functions enriched in differentially expressed gene lists from GGF-2 treated versus untreated LV tissues, identified using Ingenuity Pathway Analysis software. Functions are listed on the *y*-axis, and the *x*-axis indicates the number of differentially expressed genes in each category. Teal and orange represent low and high doses of GGF2, respectively. Enrichment *p*-value is indicated on each bar. (**b**) Pie chart showing top gene ontology categories for GGF-2 altered genes manually grouped into three generalized associative categories: development (**blue**), metabolic (**red**), and stress response (**green**). Uncharacterized and poorly characterized transcripts were excluded.

**Figure 4 biology-11-00682-f004:**
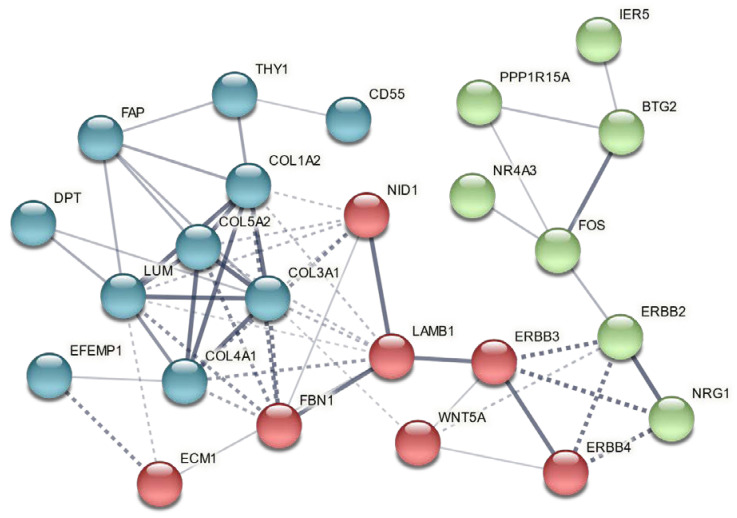
Genes commonly altered in cardiac and intercostal skeletal muscle encode functionally associated proteins. A vector graphic network created using STRING network version 11.5 with K means clustering of proteins encoded by differentially expressed genes commonly altered in both skeletal muscle and cardiac tissues of GGF2-treated post-MI pigs is shown. Full gene names with functions are provided in Table 1, except for NRG1 (neuregulin 1) and its receptors (ERBB2-4). Bubbles represent individual proteins, and lines represent associations between proteins. Line thickness indicates edge confidence, low (0.150), medium (0.400), high (0.700) and highest (0.900). Line shape indicates the predicted mode of action. Each color represents an individual cluster.

**Figure 5 biology-11-00682-f005:**
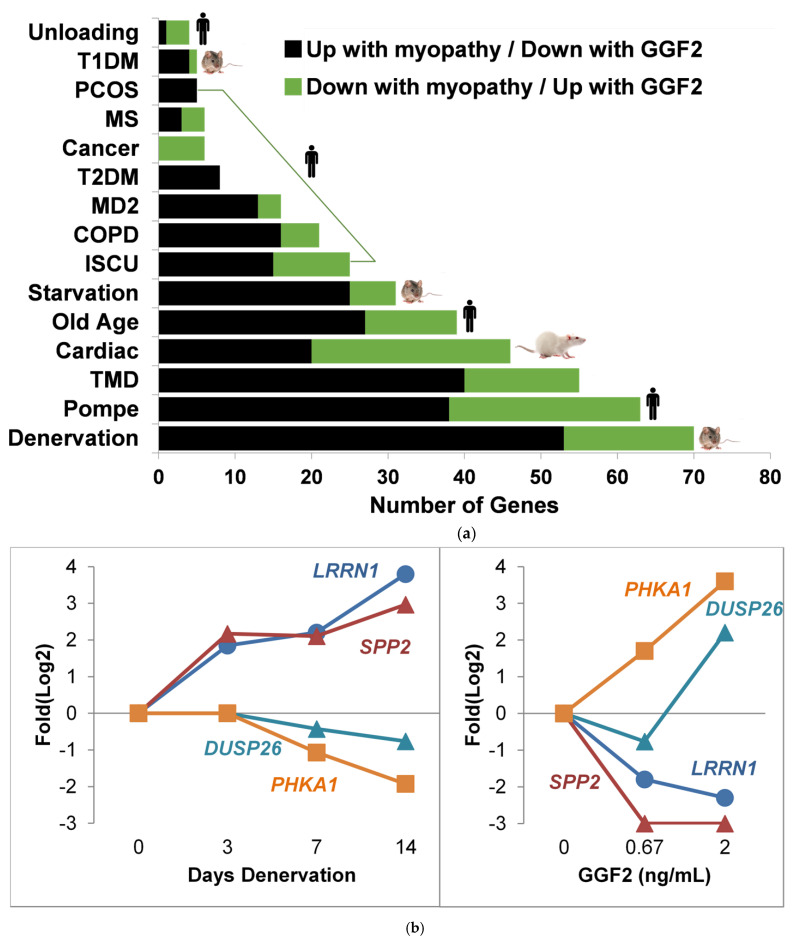
GGF2 reversibly alters myopathic genes identified through meta-analysis of repository gene expression data. (**a**) Bar chart showing numbers of genes (*x*-axis) significantly altered in various myopathies (study details listed in Table 2) that were reversibly expressed in GGF2-treated post-MI pigs. Species indicated by an icon at end of bar. Color of bar indicates directionality in listed experimental conditions relative to respective controls. Abbreviations are T1DM = (Type I Diabetes Model), PCOS = polycystic ovary syndrome, MS, T2DM = Type II diabetes mellitus, MD2 = myotonic dystrophy type II, COPD = chronic obstructive pulmonary disease, ISCU = iron-sulfur cluster scaffold homolog myopathy, and TMD = tibial muscular dystrophy. (**b**) Line graphs comparing alterations in four genes after denervation in an animal model of cachexia (left panel, GSE52676) and the same genes reversibly altered in GGF2-treated pigs (right panel). Log2 fold difference (Experiment vs. Control) is shown on the y axis, and time (in days) or dose (in ng/mL) is labeled on the *x*-axis. Colored lines correspond to genes, as labeled. LRRN1 = Leucine Rich Repeat Neuronal 1, SPP2 = Secreted Phosphoprotein 2, DUSP26 = Dual Specificity Phosphatase 26, PHKA1 = Phosphorylase Kinase Regulatory Subunit Alpha 1.

**Figure 6 biology-11-00682-f006:**
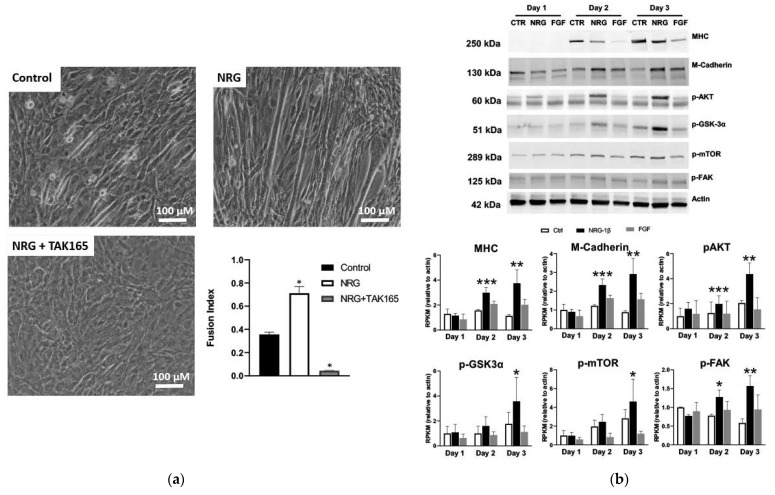
NRG-1β treatment of cultured myoblasts enhances differentiation. (**a**) C2C12 cells were cultured in differentiation media (DMEM + 2% horse serum) for 5 days in the absence (control) or presence of 10 ng/mL of recombinant neuregulin-1β (NRG, 10 ng/mL) or with both NRG and 0.2 μM of the ErbB2 receptor inhibitor TAK164. (**b**) Western blots of lysates from C2C12 cells grown in differentiation media for 5 days in the absence (control, CTR) or presence of recombinant NRG-1β (NRG) or fibroblast growth factor (FGF). Full blots provided as Supplementary Appendix A. Bar graphs, grouped by day, show results of quantification by densitometry for the indicated proteins (n = 2, n = 3 for p-FAK). Bar colors indicate treatment type: CTRL = white, NRG-1β = black, and FGF = gray. Asterisks indicate statistical significance, * *p* value < 0.05), ** *p* value < 0.005), *** *p* < 0.0005).

**Figure 7 biology-11-00682-f007:**
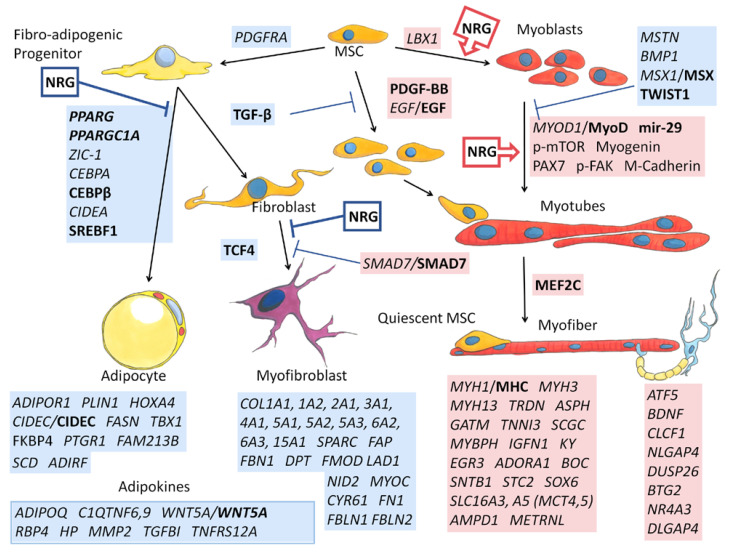
Depiction of inferred relationships of altered genes in GGF2-treated post-MI pigs. Transcripts that were up-regulated (highlighted in red) or down-regulated (blue) in intercostal muscle of neuregulin-treated pigs are italicized. Proteins identified as altered or activated based on Western blot analyses are also indicated. Bold indicates protein level inference based on functional analyses as described in the text. Known effects of neuregulin (NRG) on cell transitions based on published in vitro studies are indicated in red boxes with arrows (for promoting) or in blue (for inhibiting). Transcript abbreviations are official gene (*italic*) or protein symbols, and p- indicates phosphorylation (determined by Western blotting of lysates from NRG-treated myoblasts). MSC = mesenchymal stem cell; mir-29 = microRNA 29. Commissioned illustrations by Andrew Celso Gutierrez.

**Table 1 biology-11-00682-t001:** Genes differentially expressed in cardiac and skeletal muscle of GGF2-treated post-MI pigs.

Gene	Name	SM	LV	Function
NR4A3	Nuclear Receptor Subfamily 4 Group A Member 3	10.4	12.8	β-adrenergic inducible, Regulates transcription of fatty acid and muscle mass genes [20,21]
FOS	Fos Proto-Oncogene, AP-1 Transcription Factor Subunit	4.9	5.0	Activate phospholipid synthesis, Regulators of cell proliferation, muscle cell differentiation, and Transformation [22,23,24]
RND	Rho Family GTPase	3.6	2.6	Regulate the organization of the actin cytoskeleton in response to extracellular growth factors [54]
BTG2	BTG Anti-Proliferation Factor 2	3.3	4.1	Cell cycle regulation, cell proliferation in skeletal muscle [25]
PPP1R15A	Protein Phosphatase 1 Regulatory Subunit 15A	2.8	1.7	Down-regulates the TGF-beta, growth arrest and DNA damage-inducible protein, promoting cell death [55]
IER5	Immediate Early Response 5	2.6	4.3	Cell regulation, proliferation, and resistance to thermal stress. Dephosphorylates HSF1, and ribosomal protein S6 [56]
PAAF1	Proteasomal ATPase Associated Factor 1	2.4	1.5	Associated with Heart Conduction Disease, regulation of association of proteasome components [26,57]
ATP1B4	ATPase Na+/K+ Transporting Family Member Beta 4	2.0	1.5	Transporting protein, Transcriptional coregulator during muscle development [27,28]
COL1A2	Collagen Type 1 Alpha 2 Chain	−1.8	−2.7	Fibrillar forming collagen, putative down-regulated c-Myc target gene, or upregulate let-7b in skin fibroblasts [58].
EFEMP1	EGF Containing Fibulin Extracellular Matrix Protein 1	−1.8	−1.7	Binds EGFR receptor, autophosphorylation and the activation of downstream signaling pathways, Negative regulator of chondrocyte differentiation [59].
LAMB1	Laminin Subunit Beta 1	−1.8	−1.5	Cell adhesion, Differentiation, Encoding laminin subunit beta-1, are associated with COB with variable muscular or ocular abnormalities, Expressed in skeletal muscle [29,30]
LUM	Lumican	−2.0	−1.9	A collagen binding proteoglycan with increased expression in hearts, Regulate tissue repair, collagen fibril organization [31,32]
THY1	Thy-1 Cell Surface Antigen	−2.0	−2.6	Cell surface glycoprotein and A pathogenic CF fraction in cardiac fibrosis [33,34]
ECM1	Extracellular Matrix Protein 1	−2.1	−2.2	Response to elevated platelet cytosolic Ca2+ and ERK Signaling, Upregulated in cardiac aging and myocardial infarction [35,36]
FAP	Fibroblast Activation Protein Alpha	−2.2	−1.8	Expressed in stromal fibroblasts of epithelial cancers, tissue remodeling, healing wounds, Correlate coronary heart disease [37,38,39]
DPT	Dermatopontin	−2.3	−2.1	Extracellular matrix proteins involved in cell-matrix interaction, Postulated to modify the behavior of TGF-beta [40]
FBN1	Fibrillin 1	−2.3	−2.6	Calcium ion binding and extracellular matrix structural constituent, Differential regulation in smooth muscle cells [41,42,43]
PRPS2	Phosphoribosyl Pyrophosphate Synthetase 2	−2.3	−1.7	Phosphoribosyl pyrophosphate synthetase, protein homodimerization activity, and carbohydrate catabolic process-related genes [60,61]
NID1	Nidogen 1	−2.4	−1.8	Basement membrane glycoproteins, Laminin interactions in the heart [44,45,46]
PLSCR4	Phospholipid Scramblase 4	−2.5	−2.3	Protein coding gene; upregulated in hypertrophic mouse hearts [47,62]
WNT5A	Wnt Family Member 5A	−2.5	−1.5	Canonical and non-canonical wnt pathways, Regulating developmental pathways during embryogenesis
FSCN1	Fascin Actin-Bundling Protein 1	−2.5	−1.5	Organize F-actin; Involved in cell migration, motility, adhesion, and cellular interactions
COL4A1	Collagen Type IV Alpha 1 Chain	−2.6	−1.6	Alpha protein of Type IV collagen, components of basement membranes, Upregulated in the skeletal muscle response [48,49]
ACER3	Alkaline Ceramidase 3	−2.6	−1.5	Protein coding gene
COL5A2	Nidogen 1	−2.6	−2.5	Alpha chain for fibrillar collagen; cardiac repair and involved in Muscle-Invasive Bladder [48,50,51]
PRTFDC1	Phosphoribosyl Transferase Domain Containing 1	−2.7	−2.3	Protein Coding gene, protein homodimerization activity, and magnesium ion binding
CD55	CD55 molecule (Cromer Blood Group)	Reduced	−2.2	Glycoprotein; Regulates cell decay dysferlin is expressed in skeletal and cardiac muscles [52]
COL3A1	Collagen Type III Alpha 1 Chain	−2.8	−1.9	Fibrillar collagen found in extensible connective tissues and the vascular system [53]
HBB	Hemoglobin Subunit Beta	−3.0	−5.9	Oxygen transport from the lung, Endogenous inhibitor of enkephalin-degrading enzymes such as DPP3, and as a selective antagonist of the P2RX3 receptor which is involved in pain signaling [63]
HBA	Hemoglobin Subunit Alpha	−3.6	−5.9	Iron ion binding and oxygen transport from the lung to the various peripheral tissue [64]
ARMCX2	Armadillo Repeat Containing X-Linked 2	−3.7	−1.8	Regulate the dynamics and distribution of mitochondria in neural cells; involved in tissue development and tumorigenesis [65]

Yellow highlighted rows indicate those with relevant muscle and/or heart functions. SM = skeletal (intercostal) muscle, LV = left ventricular (cardiac muscle).

**Table 2 biology-11-00682-t002:** Gene Expression Omnibus (GEO) myopathy studies included in meta-analysis.

GEO Study ID	Description	Tissue	Species
GSE1557 [66]	Cardiac cachexia (n = 4)	Left ventricle	Rat
GSE52676 [67]	Starvation (n = 6) Denervation (n = 9) Type 1 Diabetes (n = 3)	Soleus	Mouse
GSE45331 [68]	Myotonic dystrophy type 2 (n = 6) Control (n = 4)	Vastus lat.	Human
GSE48574 [69]	ISCU (n = 3) vs. Control (n = 5)	Vastus lat.	Human
GSE38680 [70]	Pompe (n = 9) vs. Control (n = 10) Pompe (n = 11) vs. Control (n = 7)	Biceps Quad	Human
GSE34111 [71]	Cancer cachexia (n = 12) vs. Control (n = 6)	Quad	Human
GSE42806 [72]	Tibial muscular dystrophy (n = 7) vs. Healthy (n = 5)	Extensor digitorum longus	Human
GSE25941 [73]	Female: old (78 ± 1 years, n = 11) vs. young (25 ± 1 years, n = 8) Male: old (78 ± 1 years, n = 10) vs. young (25 ± 1 years, n = 7)	Vastus lat.	Human
GSE9103 [74]	Old (n = 65.1 ±1.5, n = 10) vs. young (22.7 ± 0.7, n = 10)	Vastus lat.	Human
GSE5110 [75]	48 h immobilization vs. control: male subjects (n = 5)	Vastus lat.	Human
GSE21496 [76]	48 h suspension vs. control: sedentary male subjects (n = 7)	Vastus lat.	Human
GSE43760 [77]	Metabolic syndrome (n = 6) vs. healthy (n = 6)	Vastus lat.	Human
GSE27536	COPD low BMI (n = 6) vs. healthy (n = 12) COPD normal BMI (n = 8) vs. healthy (n = 12)	Vastus lat.	Human
GSE6798 [78]	Obese + PCOS (n = 16) vs. Control (n = 13)	Vastus lat.	Human
GSE8157 [79]	Obese + PCOS (n = 10) vs. Control (n = 13)	Vastus lat.	Human
GSE19420	Type 2 diabetes (n = 10) vs. normoglycemic subjects (n = 12)	Vastus lat.	Human
GSE25462 [80]	Type 2 diabetes (n = 10) vs. normoglycemic + no family history of diabetes (n = 15) Type 2 diabetes (n = 10) vs. normoglycemic + family history of type 2 diabetes (n = 25)	Quad	Human

**Table 3 biology-11-00682-t003:** Predicted upstream regulators for GGF2-altered Genes.

Upstream Regulator	Gene Expression	No. Downstream Targets Altered (*p*-Value)	Z Score (Predicted State)
Transforming growth factor β1 (TGF-β1)	-	114 (1.6 × 10^−23^)	Z = −2.297 (Inhibited)
Platelet derived growth factor-BB (PDGF BB)	-	36 (3.7 × 10^−22^)	Z = 1.567
Angiotensinogen (AGT)	−3.8	67 (8.5 × 10^−20^)	Z = −3.032 (Activated)
Fibroblast growth factor 2 (FGF2)	-	45 (4.8 × 10^−20^)	nd
CAMP Responsive Element Binding Protein 1 (CREB1)	-	46 (3.5 × 10^−19^)	Z = 2.424 (Activated)
Erb-B2 Receptor Tyrosine Kinase 2 (ERBB2)	-	60 (5.7 × 10^−19^)	nd
PPARG coactivator 1 α (PCG-1α)	-	37 (4.4 × 10^−18^)	Z = 1.877
Aryl hydrocarbon receptor (AHR)	-	45 (4.5 × 10^−17^)	Z = 3.63 (Activated)
Twist family BHLH transcription factor 1 (TWIST1)	-	23 (6.7 × 10^−17^)	Z = −2.449 (Inhibited)
Epidermal growth factor (EGF)	3.5	49 (5.3 × 10^−16^)	nd
Cadherin associated protein α1 (α-catenin)	-	19 (3.0 × 10^−15^)	Z = 3.118 (Activated)
Erb-B3 Receptor Tyrosine Kinase 3 (ERRB3)	-	18 (2.5 × 10^−13^)	nd
Mothers against DPP homolog 7 (SMAD7)	2.4	18 (3.3 × 10^−12^)	Z = 3.11 (Activated)
CCAAT enhancer binding protein β (C/EBP-β)	-	37 (1.1 × 10^−11^)	Z = −2.091 (Inhibited)
AKT serine/threonine kinase 1 (AKT1)	-	26 (1.9 × 10^−11^)	Z = −1.855
Enalapril (Hypertension medication)	-	13 (9.3 × 10^−11^)	Z = 3.073 (Activated)
CCAAT enhancer binding protein α (C/EBP-α)	−9.3	28 (2.5 × 10^−11^)	Z = −1.812
Brain-derived neurotrophic factor (BDNF)		30 (6.7 × 10^−11^)	nd
Transforming growth factor β3 (TGF-β3)	-	17 (7.6 × 10^−11^)	Z = −1.937
Myocilin (MYOC)	−2.5	13 (1.7 × 10^−10^)	nd
Myocyte enhancer factor 2C (MEF2C)	-	15 (3.7 × 10^−10^)	Z = 2.912 (Activated)
Peroxisome proliferator activated receptor γ (PPAR-γ)	−5.4	36 (8.6 × 10^−10^)	Z = −1.987
Sterol regulatory element-binding transcription factor 1 (SREBF1)	-	20 (2.4 × 10^−9^)	Z = −1.937
Erb-B3 Receptor Tyrosine Kinase 3 (ERRB4)	-	12 (3.4 × 10^−8^)	Z = 2.388 (Activated)
Transforming growth factor β1 (TGF-β1)	-	15 (4.3 × 10^−8^)	Z = −1.634
microRNA-29b-3p (miR-29b-3p)	-	13 (4.8 × 10^−8^)	Z = 3.097 (Activated)
Thapsigargin (Calcium reuptake inhibitor)	-	20 (6.5 × 10^−8^)	Z = 3.450 (Activated)
microRNA-335-3p (miR-335-3p)	-	8 (6.7 × 10^−8^)	Z = 2.828 (Activated)
CCAAT enhancer binding protein δ (C/EBP-δ)	-	13 (1.3 × 10^−7^)	Z = −1.813
Peroxisome proliferator activated receptor α (PPAR-α)	-	30 (2.5 × 10^−7^)	Z = −1.363
Clopidogrel (antiplatelet blood-thinning medication)		11 (4.6 × 10^−7^)	Z = 3.302 (Activated)
microRNA lethal 7a-5p (Let-7a-5p)	-	17 (1.4 × 10^−6^)	Z = 3.682 (Activated)
Myocardin (MYOCD)	-	9 (4.4 × 10^−6^)	Z = 2.759 (Activated)
Cell death inducing DFFA like effector C (CIDEC)	−9.4	6 (1.1 × 10^−5^)	nd
Msh homeobox 1 (MSX1)	−1.7	5 (1.8 × 10^−5^)	Z = −1.982
microRNA-30c-5p (miR-30c-5p)	-	17 (2.1 × 10^−5^)	Z = 3.117 (Activated)
sterol regulatory element binding transcription factor 2 (SREBF2)	-	9 (5.8 × 10^−5^)	Z = −2.394 (Inhibited)
microRNA-21 (miR-21)	-	17 (3.3 × 10^−5^)	Z = −2.668 (Inhibited)
Myogenic differentiation 1 (MYOD1)	2.2	16 (3.7 × 10^−5^)	nd
Wnt family member 5a (WNT5a)	−2.5	10 (8.8 × 10^−5^)	Z = −2.394 (Inhibited)
Peroxisome proliferator activated receptor δ (PPAR-δ)	-	15 (8.5 × 10^−4^)	Z = −1.214
26s Proteosome (protein complex)	-	11 (5.5 × 10^−4^)	Z = −2.035 (Inhibited)
Transcription factor 4 (TCF4)	-	18 (8.7 × 10^−4^)	Z = −1.554
PPARG coactivator 1 β (PCG-1β)	-	5 (1.6 × 10^−2^)	Z = −2.186 (Inhibited)

## Data Availability

Original study data are available in the Gene Expreassion Omnibus under Accession series number GSE48255 (https://www.ncbi.nlm.nih.gov/bioproject/PRJNA209383). Data were accessed June 2017 and February 2022.

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
