# Peer review of "Neuregulin (NRG-1β) Is Pro-Myogenic and Anti-Cachectic in Respiratory Muscles of Post-Myocardial Infarcted Swine"

_biology, 2022, doi:10.3390/biology11050682_

Round 1
Reviewer 1 Report
In this study, the authors perform deep sequencing analysis on intercostal muscles of a swine model of heart failure treated minus or plus a pharmaceutical-grade version of GGF2 to determine the effects of GGF2 treatment on post-MI skeletal muscle function and compared these results with GGF2-mediated expression changes in cardiac muscle and myopathic genes obtained by analyzing repository expression data. They show that GGF2 treatment of these heart failure animals alter a plethora of intercostal muscle gene expression transcripts, several associated with potentially attenuating skeletal muscle cachexia. Overall, this is a well written manuscript with an impressive amount of data in a clinically relevant model which will serve as a good resource for other investigators. The link to an involvement of a GGF2-induced skeletal muscle signaling pathway is interesting and Figure 7 is particularly informative.
Major comments:
- It is unfortunate that the authors did not test whether the GGF2-mediated improvement in gene expression profile could be linked or at least correlated to overt functional improvements in these heart failure animals related to skeletal muscle biology. For example, the authors could provide experimental evidence for improvement of dyspnea. Alternatively, if the data is not available, this limitation should be highlighted in the revised Discussion as a potential future study.
- The animals are supplemented i.v. with GGF2. Have the authors also considered that a potential improvement in overall function might additionally involve central events? This could be included in the Discussion.
- Only two animals received the high dose of GGF2. Statistical validation should not be implied for these animals.
Minor comments
Line 51… that induce
Author Response
Responses to Reviewer 1 Comments
Reviewer 1: Overall, this is a well written manuscript with an impressive amount of data in a clinically relevant model which will serve as a good resource for other investigators. The link to an involvement of a GGF2-induced skeletal muscle signaling pathway is interesting and Figure 7 is particularly informative.
Response: We thank the reviewer for taking the time to assess our manuscript and for providing valuable comments.
Major comment 1: It is unfortunate that the authors did not test whether the GGF2-mediated improvement in gene expression profile could be linked or at least correlated to overt functional improvements in these heart failure animals related to skeletal muscle biology. For example, the authors could provide experimental evidence for improvement of dyspnea. Alternatively, if the data is not available, this limitation should be highlighted in the revised Discussion as a potential future study.
Response: Unfortunately we did not measure respiratory muscle function or any related measure. We have added a sentence to the Discussion indicating the need for further work to address the potential clinical implications of this work.
Major comment 2: The animals are supplemented i.v. with GGF2. Have the authors also considered that a potential improvement in overall function might additionally involve central events? This could be included in the Discussion.
Response: This is an interesting concept that deserves some further investigation and discussion, that relates more to the cardiac effects reported in our prior manuscript. There did not appear to be a good place to advance this idea in the current manuscript, but if the reviewer thinks differently please let us know.
Major comment 3: Only two animals received the high dose of GGF2. Statistical validation should not be implied for these animals.
Response: We agree that more animals in this category would be preferable. However, this was not possible due to issues with tolerability due to symptomatic hypoglycemia. Which per IACUC precluded us from adding more animals to this group. This has been added to the methods section.
Minor comment 1: Line 51… that induce
Response: We thank the reviewer for finding this typo and have made the correction. We have also re-read the manuscript carefully to ensure correctness.
Reviewer 2 Report
the first and second sentences of the intro are quite disconnected. The authors should improve on that.
My major concern relates to how results are shown in the article. For example, I am not convinced gene expression data in figure 2 is shown in an appropriate fashion. Genes should not be connected as they're individual genes. If the authors want to show a selected group of differentially-expressed genes, they should use a reduced heat-map or plot expression levels of each animal separately with standard dev. Also using only 2 animals for the high dose is tricky, no standard dev can be calculated, reducing the significance of the work.
For figure 6, what is the n of that experiment? How many independent cultures have been done? Please provide statistical information.
It would be nice to see quantification graphs of western blot in figure 6. Ideally as scatter plot.
Table 3 legend doesn't make sense.
Minor edits for typos, for instance, line 51: 'thatinduce' should be 'that induce'
Please explain LV and SM in the table.
Author Response
Responses to Reviewer 2 Comments
Reviewer 2: My major concern relates to how results are shown in the article.
Response: We appreciate the reviewer’s time and comments, which we believe allowed us to substantially improve the manuscript accordingly.
Major Concern 1: For example, I am not convinced gene expression data in figure 2 is shown in an appropriate fashion. Genes should not be connected as they're individual genes. If the authors want to show a selected group of differentially expressed genes, they should use a reduced heat-map or plot expression levels of each animal separately with standard dev.
Response: Per reviewer suggestions, we replaced the previous plots with Dot plots showing relative counts (as RPKM) for samples grouped by treatment (generated using Partek Genomics Suite).
Major Concern 2: Also using only 2 animals for the high dose is tricky, no standard dev can be calculated, reducing the significance of the work.
Response: We agree that more animals in this category would be preferable. However, this was not possible due to issues with tolerability due to symptomatic hypoglycemia. Which per IACUC precluded us from adding more animals to this group. This has been added to the methods section.
Major Concern 3: For figure 6, what is the n of that experiment? How many independent cultures have been done? Please provide statistical information.
Response: We provided the requested information in the figure legend.
Major Concern 4: It would be nice to see quantification graphs of western blot in figure 6. Ideally as scatter plot.
Response: We provided the densitometry for Figure 6, although given the small number of pure replicates felt it best to show as bar graphs. That said, the experiments were performed in two different cell lines, two rodent species, and with several versions of NRG, including GGF2 (as detailed in the additional methods, Appendix A). Whereas similar in vitro studies of NRG using these cell lines has been performed, the novelty here biological replication that does not provide the same level of statistical power as technical replicates but does provide a much higher level of biological confidence. We hoped to show that the aggregate of the data support the physiological consequences implied by altered transcriptional and protein levels.
Major Concern 5: Table 3 legend doesn't make sense.
Response: We have removed the legend, which we included as superfluous information showing which microarrays were used for each study. However, this information is somewhat redundant given that these details are available on the repository website (and also do not really add much here).
Major Concern 6: Minor edits for typos, for instance, line 51: 'that induce' should be 'that induce'
Response: We thank the reviewer for finding this typo and have made the correction. We have also re-read the manuscript carefully to ensure correctness.
Major Concern 7: Please explain LV and SM in the table.
Response: We provided the requested information in the Table legend as follows: “SM = skeletal (intercostal) muscle, LV = Left Ventricular (cardiac muscle)”
Other Concern: The first and second sentences of the intro are quite disconnected. The authors should improve on that.
Response: To address this issue and hopefully provide a more natural flow, we added the following connector phrase at the beginning of the second sentence: “Although the precise mechanisms are not fully understood, some studies recently showed that system…”

Round 2
Reviewer 2 Report
The work has improved, particularly fig 2.
Author Response
Thank you for your helpful comments to improve our manuscript!